# Lipopolysaccharide-Enhanced Responses against Aryl Hydrocarbon Receptor in FcgRIIb-Deficient Macrophages, a Profound Impact of an Environmental Toxin on a Lupus-Like Mouse Model

**DOI:** 10.3390/ijms22084199

**Published:** 2021-04-18

**Authors:** Kanyarat Udompornpitak, Thansita Bhunyakarnjanarat, Awirut Charoensappakit, Cong Phi Dang, Wilasinee Saisorn, Asada Leelahavanichkul

**Affiliations:** 1Medical Microbiology, Interdisciplinary and International Program, Graduate School, Chulalongkorn University, Bangkok 10330, Thailand; jubjiibb@hotmail.com; 2Department of Microbiology, Faculty of Medicine, Chulalongkorn University, Bangkok 10330, Thailand; 3Translational Research in Inflammation and Immunology Research Unit (TRIRU), Department of Microbiology, Chulalongkorn University, Bangkok 10330, Thailand; thansitadew@gmail.com (T.B.); awirut.turk@gmail.com (A.C.); pilotdang1308@gmail.com (C.P.D.); wsaisorn@gmail.com (W.S.)

**Keywords:** FcgRIIb-deficient mice, systemic lupus erythematosus, aryl hydrocarbon receptor, air pollution

## Abstract

Fc gamma receptor IIb (FcgRIIb) is the only inhibitory-FcgR in the FcgR family, and FcgRIIb-deficient (FcgRIIb^−/−^) mice develop a lupus-like condition with hyper-responsiveness against several stimulations. The activation of aryl hydrocarbon receptor (Ahr), a cellular environmental sensor, might aggravate activity of the lupus-like condition. As such, 1,4-chrysenequinone (1,4-CQ), an Ahr-activator, alone did not induce supernatant cytokines from macrophages, while the 24 h pre-treatment by lipopolysaccharide (LPS), a representative inflammatory activator, prior to 1,4-CQ activation (LPS/1,4-CQ) predominantly induced macrophage pro-inflammatory responses. Additionally, the responses from FcgRIIb^−/−^ macrophages were more prominent than wild-type (WT) cells as determined by (i) supernatant cytokines (TNF-α, IL-6, and IL-10), (ii) expression of the inflammation associated genes (*NF-κB, aryl hydrocarbon receptor*, *iNOS, IL-1β* and activating-*FcgRIV*) and cell-surface CD-86 (a biomarker of M1 macrophage polarization), and (iii) cell apoptosis (Annexin V), with the lower inhibitory-*FcgRIIb* expression. Moreover, 8-week-administration of 1,4-CQ in 8 week old FcgRIIb^−/−^ mice, a genetic-prone lupus-like model, enhanced lupus characteristics as indicated by anti-dsDNA, serum creatinine, proteinuria, endotoxemia, gut-leakage (FITC-dextran), and glomerular immunoglobulin deposition. In conclusion, an Ahr activation worsened the disease severity in FcgRIIb^−/−^ mice possibly through the enhanced inflammatory responses. The deficiency of inhibitory-FcgRIIb in these mice, at least in part, prominently enhanced the pro-inflammatory responses. Our data suggest that patients with lupus might be more vulnerable to environmental pollutants.

## 1. Introduction

Aryl hydrocarbon receptor (Ahr), a cytosol receptor, [1,2,3,4,5,6] is a natural sensor for either the exogenous factors, including polyaromatic hydrocarbons and environmental toxins, or the oxygen-related endogenous factors [7]. Indeed, several pollutants activate Ahr and initiate pathogenic inflammatory responses. For example, polycyclic aromatic hydrocarbons (PAHs) in the particulate matter of air pollution [8,9] induce lung injury and systemic inflammation [10] through Ahr activation [11,12,13] are demonstrated in several studies [1,3]. The dietary components, indole groups in some vegetables, and some amino acids (tryptophan and cysteine) could be metabolized, partly by gut microbiota, into several Ahr agonists [7]. Nevertheless, Ahr activation produces both pro- [11] and anti-inflammatory effects [1,3,14,15] possibly depending on the cell microenvironment. Interestingly, inflammation enhances the pollutant effects [16], and the pollutants also amplified inflammatory responses [7]. As such, Ahr signaling is escalated in macrophages that are pre-conditioned by mycobacterial infection [17] and the exposure to PAHs accelerates serum inflammatory markers [11]. Hence, there might be an individual difference in the response against Ahr activation among persons due to the variations in genes and epigenetics [18]. Interestingly, macrophages are major innate immune cells that are responsible for the pollutants-induced inflammation, possibly due to their professional phagocytosis activity and cytokine production [19,20,21]. Among phagocytic cells (macrophages, neutrophils, and dendritic cells) [22,23], macrophages (referred to as “sentinel immune cells”) initiate inflammation for the subsequent healing processes [24,25] and are distributed throughout the body [26,27]. Despite the various forms of Ahr agonists with several routes of contamination, these substances could induce macrophages in several parts of the body, including liver (Kupffer cells), brain (microglia), kidney (mesangial cells), lung (alveolar macrophages), and intestine (intestinal macrophages) [28]. 

In parallel, lupus or systemic lupus erythematosus (SLE) is a common systemic autoimmune disease caused by multifactorial pathogenesis including several genetic defects and environmental factors. A high prevalence of the dysfunction polymorphism in Fc gamma receptor IIb (FcgRIIb), the only inhibitory receptor in FcgR family, among Asians is mentioned [29,30,31,32]. In addition, FcgRIIb-deficient (FcgRIIb^−/−^) mice demonstrate several characteristics of lupus (lupus-like condition) and have been used as a representative lupus model with the spontaneous development of anti-dsDNA, an important lupus autoantibody, as early as 16–24 weeks of age [33,34,35]. The defects of inhibitory-FcgRIIb in B cells and plasma cells, which enhances production of autoantibodies and circulating immune complexes (CICs), is a possible major mechanism of spontaneous development of lupus characteristics in some patients [31,36,37,38]. On the other hand, the defects of FcgRIIb in macrophages improve bactericidal activity against malaria in a human study [39] implying a possible macrophages hyper-activity due to the loss of inhibitory FcgRIIb signals. Additionally, the environmental factors, including silica, current cigarette smoking, air pollution, ultraviolet light, solvents, pesticides, and heavy metals, increase risk of SLE [40,41,42]. These data highlight an association between environment and lupus activity.

Likewise, the inhibitory loss in FcgRIIb^−/−^ mice induces immune hyper-responsiveness, not only against self-antigens but also toward pathogen molecules [31,34,43,44,45]. As such, lipopolysaccharide (LPS), a major component of Gram-negative bacterial cell wall, is one of the endogenous pathogen molecules due to the high abundance of Gram-negative bacteria in gut. In active lupus, it is possible that the deposition of CICs induces the intestinal mucosal injury [46] and initiates gut translocation of endotoxin that worsens lupus activity through systemic inflammation [34,47]. The detection of endotoxemia in patients with active lupus [46] supports a possible clinical importance of LPS and gut permeability defect in lupus. Because several environmental factors [48] exacerbate lupus activity and several pollutants enhance inflammation [49,50,51], patients with lupus might be more vulnerable to pollutants than healthy people. Indeed, some environmental protections, including avoidance from ultraviolet and heat, are commonly recommended for the patients with lupus [48]. Currently, there are more severe environmental problems with several emerging pollutants that might also affect patients with lupus. The recommendation in wearing a face mask or avoiding some diets to protect from air pollutants or diet-derived Ahr agonists, respectively, might need to be emphasized in patients with lupus. 

More importantly, an impact of environmental factors on the undiagnosed lupus-prone condition is even more interesting as pollutants might induce new cases of lupus in the persons with some genetic variations and the environmental controls might be useful for the prevention. Due to the age dependency in development of lupus-like characteristics in FcgRIIb^−/−^ mice, 8 week old FcgRIIb^−/−^ mice are a representative model of asymptomatic lupus-prone mice without any signs of lupus characteristics [33,34,35]. On the other hand, the lupus-like characteristics in FcgRIIb^−/−^ mice are presented as early as 16–24 weeks old (increased anti-dsDNA) and full-blown lupus nephritis (proteinuria, elevated serum creatinine, and glomerular CIC deposition) at 32–40 weeks old [33,34,35]. Here, we hypothesized that the environmental pollutants, as represented by an Ahr activator, might affect persons with lupus-prone genes more prominently then the healthy persons. Then, we tested the hypothesis in mice and in macrophages from FcgRIIb^−/−^ and wild-type (WT) groups.

## 2. Results

The pre-treatment with LPS amplified expression of aryl hydrocarbon receptor (Ahr) resulted in the more profound responses of FcgRIIb^−/−^ macrophages against the subsequent activation by Ahr agonist, a representative environmental toxin, when compared with WT cells. These activations led to the more severe inflammation that activated lupus-like characteristics in FcgRIIb^−/−^ mice.

### 2.1. Pre-Treatment with LPS Enhanced Macrophage Responses toward Ahr Activator, But Not Vice Versa, in the Macrophage Cell Line

Due to the possible differences in responses toward Ahr agonist with or without LPS pre-conditioning, RAW264.7 cells (a macrophage cell line) were activated by Ahr agonist, alone or with LPS (see Section 4 Materials and Methods and Figure 1A). As such, Ahr activation alone using 1,4-CQ (N/1,4-CQ) (see Figure 1A) did not induce any inflammatory responses in macrophages as determined by supernatant cytokines and the expression of several genes (Figure 1B–J and Figure 2A–F). Meanwhile, a single stimulation by LPS, a potent inflammatory activator, (N/LPS) induced profound macrophage responses but did not differ from LPS stimulation after 1,4-CQ pre-treatment (1,4-CQ/LPS) (Figure 1B–J and Figure 2A–F). In parallel, LPS pre-treatment before 1,4-CQ (LPS/1,4-CQ), when compared with Ahr alone (N/1,4-CQ), enhanced several inflammatory markers, including supernatant cytokines and gene expression of *TLR-4*, *iNOS,* and *IL-1β* (Figure 1B–J and Figure 2A–F). Notably, N/LPS was evaluated at 24 h post LPS, while, in LPS/1,4-CQ, the samples were collected at 54 h post LPS stimulation with a 6 h wash-out procedure (Figure 1A). These difference in the treatment duration was responsible for the low supernatant cytokines at the 0 time-point in LPS/1,4-CQ group (Figure 1B–D). The comparison between LPS/1,4-CQ and N/1,4-CQ represents the cells with pre-stimulation versus the regular cells without previous activation, respectively. Interestingly, LPS activation (N/LPS) enhanced *Ahr* expression as early as 2 h after the activation (Figure 1J) which might be responsible for the vigorous responses against Ahr agonist after LPS (LPS/Ahr). The activation by all of these protocols in RAW246.7 cells did not induce anti-inflammatory M2 macrophage polarization as determined by gene expression of *Arginase* and *TGF-β* (Appendix A). 

### 2.2. Pre-Treatment with LPS before Ahr Activation in FcgRIIb^−/−^ Macrophages Induced More Severe Inflammation Than WT Cells

The experiments on RAW264.7 cells demonstrated the different responses toward Ahr agonist alone (N/1,4-CQ) and with LPS pre-treatment (LPS/1,4-CQ). Then, bone marrow-derived macrophages were further used to explore the difference between FcgRIIb^−/−^ versus WT cells. With LPS stimulation alone (N/LPS), FcgRIIb^−/−^ macrophages demonstrated more potent inflammatory responses than WT cells as indicated by the higher level of several indicators (at least in one time-point), including supernatant cytokines (Figure 3A–C, left side) and the gene expression (Appendix A), the downstream inflammatory signals (*TLR-4* and *Ahr*) (Figure 3D–F, left side), the M1 polarization associated genes (*iNOS* and *IL-1β*), and a surface marker (CD86 by flowcytometry analysis) (Figure 4A,B, left side and Figure 4C). Interestingly, both inhibitory-*FcgRIIb* and activating-*FcgRIV* were upregulated in WT macrophages, while only activating-*FcgRIV*, but not *FcgRIIb*, were enhanced in FcgRIIb^−/−^ macrophages after N/LPS stimulation (Figure 4D–F, left side). There was no expression of *FcgRI* in all conditions (data not shown) and no expression of M2 polarization-associated markers (*Arginase*, *TGF-β* and CD206 surface marker) at 24 h post-stimulation (Appendix A, G at left side and Appendix A). The representative pictures of flow-cytometry analysis are demonstrated in Appendix A. These data supported the LPS hyper-responsiveness of FcgRIIb^−/−^ macrophages as previously mentioned [43,52]. In parallel, macrophage responses against 1,4-CQ before LPS stimulation (1,4-CQ/LPS) was similar to the responses toward LPS alone (N/LPS) with the more prominent responses in FcgRIIb^−/−^ macrophages than WT cells after 1,4-CQ/LPS and N/LPS stimulation (Figure 3 and Figure 4, right side). The pre-treatment with 1,4-CQ before LPS (1,4-CQ/LPS) showed a neutral effect in comparison with LPS alone (N/LPS). Meanwhile, Ahr activator alone (N/1,4-CQ) did not activate any of these indicators (Figure 3 and Figure 4, left side). Hence, Ahr activation alone (N/1,4-CQ) demonstrated no effect on macrophages, and the pre-treatment with Ahr agonist before LPS (1,4-CQ/LPS) did not enhance the responses compared with LPS alone (N/LPS). 

However, LPS pre-treatment before Ahr stimulation (LPS/1,4-CQ) in WT macrophages showed a slight elevation of supernatant IL-6 (Figure 3B, right side) and *IL-1β* expression (Figure 4B, right side) (compared with the baseline) and increased CD86 (compared with the control group) (Figure 4C). In LPS/1,4-CQ on FcgRIIb^−/−^ macrophages, nearly all of the pro-inflammatory parameters were enhanced with a higher level than LPS/1,4-CQ stimulated WT cells, including supernatant cytokines with the gene expression (TNF-α, IL-6 and IL-10), the downstream molecular signals (*NF-κB* and *Ahr*), M1 macrophage polarization (*iNOS*, *IL-1β* and CD86), and *FcgRIV* expression (Figure 3 and Figure 4, right side). Notably, LPS/1,4-CQ altered FcgRIIb^−/−^ macrophages into the pro-inflammatory M1 polarization as indicated by all biomarkers of M1 polarization (*iNOS*, *IL-1β*, and CD86) while LPS/1,4-CQ increased only *IL-1β* and CD86, but not *iNOS*, (Figure 4A–C, right side) implied a lesser effect of LPS/1,4-CQ on WT macrophages compared with FcgRIIb^−/−^ cells.

Interestingly, there was no expression of inhibitory *FcgRIIb*, but increased expression of the activating *FcgRs* (*FcgRIII* and *FcgRIV*) in FcgRIIb^−/−^ macrophages (when compared with the baseline) after the stimulations (Figure 4D–F). Perhaps, the hyper-responsiveness in FcgRIIb^−/−^ macrophages might be responsible from the absence of inhibitory FcgRIIb with the enhanced activating FcgRs. Despite a neutral effect on the stimulation by Ahr agonist alone, LPS pre-treatment enhanced responses to Ahr agonist (LPS/1,4-CQ) in macrophages from both mouse strains with the more prominent activity in FcgRIIb^−/−^ macrophages (Figure 3 and Figure 4, right side). These data indicated that priming with pro-inflammatory activators (by LPS) accelerated the responses against an environmental toxin (Ahr agonist). The subsequent Ahr stimulation in LPS/1,4-CQ might be severe enough to enhance cell injury similar to the enhanced-cell damage from the pre-conditioning inflammation as previously published [53,54,55]. Accordingly, LPS/1,4-CQ, but not 1,4-CQ alone, in FcgRIIb^−/−^*,* macrophages induced higher abundance of the late apoptosis cells when compared with WT cells (Figure 5A–D). Because cell apoptosis is a lupus exacerbating factor [51], the easier apoptosis induction in FcgRIIb^−/−^ cells with Ahr activation implied a possible consequence of the enhanced apoptosis in the exacerbation of lupus-like condition in FcgRIIb^−/−^ mice.

### 2.3. Prominent Inflammatory Activation and the Exacerbation of Lupus-Like Condition after the Activation of Aryl Hydrocarbon Receptor in FcgRIIb^−/−^ Mice

It is well-known that lupus activity is exacerbated by inflammation [49,50]. Hence, 1,4-CQ was intraperitoneally administered in 8 week old FcgRIIb^−/−^ and WT mice in the short and the long-term protocols to explore the inflammatory activity of an Ahr agonist. In the short-term administration, Ahr activator alone (PBS/1,4-CQ) (see schema in Figure 6) did not induce inflammatory cytokines in both FcgRIIb^−/−^ and WT mice (Figure 6A–C). Meanwhile, inflammation in FcgRIIb^−/−^ mice after 24 h of LPS activation (LPS/PBS) was slightly higher than WT mice as indicated by TNF-α and IL-10 (at 0 h time-point) (Figure 6A,C). Nevertheless, LPS pre-treatment in mice before Ahr agonist (LPS/1,4-CQ) enhanced the responses against Ahr activator, when compared with Ahr agonist alone (PBS/1,4-CQ), in both WT and FcgRIIb^−/−^ mice (Figure 6A–C). There was non-responsiveness in 1,4-CQ activation alone (PBS/1,4-CQ) in both mouse strains and all serum cytokines of LPS/1,4-CQ activated FcgRIIb^−/−^ mice were more predominant than WT mice at 2 h and 6 h of the stimulation (Figure 6A–C). 

Although there was a limited response of FcgRIIb^−/−^ mice in a single stimulation by Ahr activator, a long-term administration might be different, considering the vulnerability against several stimuli in patients with lupus [42]. Accordingly, the once daily intraperitoneal 1,4-CQ injection for 8 weeks (Figure 7 schema) induced anti-dsDNA, proteinuria, and serum cytokines (but not increased serum creatinine) only in FcgRIIb^−/−^ mice (not in WT mice) (Figure 7A–F).

In addition, the systemic inflammation from Ahr activation in FcgRIIb^−/−^ mice was severe enough to induce gut permeability defect (gut leakage) as indicated by FITC-dextran assay and endotoxemia (Figure 7G,H). Moreover, the inflammation from endotoxemia and cytokinemia exacerbated lupus activity as demonstrated by accelerated glomerular CIC deposition (immunofluorescence) (Figure 7I,J), renal inflammation (cytokines in renal tissue) and glomerular mesangial expansion by H&E staining (Figure 8A–E). Of note, the high-magnification H&E-stained picture indicates the larger size of glomeruli in 1,4-CQ FcgRIIb^−/−^ mice compared with 1,4-CQ WT mice (Figure 8E). In addition, the injury in other parts of kidneys (tubules, interstitial area, and vessels) with 1,4-CQ administration was non-different from the PBS control group (in both mouse strains) (data not shown). As such, the glomerular immune complex deposition with a subtle change in glomerular histology is similar to Class I lupus nephritis [56]. In PBS administered FcgRIIb^−/−^ mice (control), there was no characteristic of the lupus-like condition despite the increased age of the FcgRIIb^−/−^ mice (Figure 7 and Figure 8).

## 3. Discussion

Here, LPS upregulated *Ahr* expression which enhanced the responses toward a subsequent stimulation by Ahr activator, a representative environmental toxin, more prominently in FcgRIIb^−/−^ macrophages when compared with WT cells. The long-term Ahr administration in asymptomatic FcgRIIb^−/−^ mice induced the activity of the lupus-like condition as indicated by serum anti-dsDNA and glomerular CIC deposition suggested a possible adverse effect of environmental toxins on lupus.

### 3.1. The Enhanced Macrophage Responses against an Aryl Hydrocarbon Receptor Activator by a Pre-Conditioning Immune Activation, an Impact of Inflammation from the Environmental Toxins

Although LPS was primarily selected to be used as a representative potent pro-inflammatory substance, LPS could be presented in serum due to the translocation from gut into blood circulation (leaky gut) [57]. Leaky gut could be presented with or without gastrointestinal symptoms (abdominal pain and diarrhea) [58,59], but it is commonly associated with colitis due to infections [58,59] or non-infectious causes [60,61]. Accordingly, non-colitis endotoxemia in humans and mouse models are mentioned, including (i) metabolic endotoxemia in obesity and vigorous exercise [62,63], (ii) drugs and diets [47,64], and (iii) sepsis [61,65]. In patients with lupus, endotoxemia is presented during an inactive and active disease activity [34] suggesting a possible impact of LPS in lupus activity and the flare-up. An influence of environmental stimulations in patients with or without endotoxemia might be different. While Ahr activator alone (N/1,4-CQ) did not activate macrophages, the sequential stimulation with LPS followed by Ahr activator (LPS/1,4-CQ) enhanced the responses implied a possible impact of inflammation before environmental stimulation.

In LPS/1,4-CQ macrophages (FcgRIIb^−/−^ and WT), increased several inflammatory markers (from the baseline) as early as 3–6 h including pro-inflammatory cytokines, inflammatory downstream signals (*TLR-4*, *Ahr* and *NF-κB*), M1 macrophage polarization (*iNOS*, *IL-1β* and CD86), and activating-FcgRs (*FcgRIII* and *FcgRIV*) were demonstrated. However, the inhibitory-*FcgRIIb* was upregulated only in LPS/1,4-CQ macrophages of WT, but not FcgRIIb^−/−^ cells, perhaps for an anti-inflammatory balance. In FcgRIIb^−/−^ macrophages, only the activating-*FcgRs*, but not the inhibitory-*FcgRIIb*, were detectable which might enhance the TLR-4 responses (TLR-4 and FcgRs crosstalk) [66] and a shift of balance toward profound inflammation [67]. In parallel, Ahr agonist also accelerates TLR-4 activation [68]. These mechanisms might be responsible for the enhanced responses to Ahr activator after LPS pre-treatment in FcgRIIb^−/−^ macrophages. In contrast, pre-treatment by Ahr activator before LPS did not alter LPS responses implied a low inflammatory property of Ahr agonist alone and highlighted an importance of other compounds that combined with Ahr agonist in the environmental toxins. For example, Ahr agonist of the particulate matter (PM) in air pollution (a hydrocarbon main structure) might be less toxic to the patients but the other attached components (heavy metals, nitrate, sulfate, and ammonium) might be a main cause of toxicity [14,69,70]. Because of the possible spontaneous endotoxemia in patients with lupus [34,46], our data implied an impact of the condition that initiate inflammation, especially leaky-gut-induced endotoxemia, in a lupus-like condition before the stimulation by environmental toxins [71,72,73]. The avoidance of endotoxemia should also be considered in patients with lupus.

### 3.2. Prominent Inflammatory Responses in FcgRIIb^−/−^ Macrophages over the Wild-Type Cells, an Inhibitory Effect of FcgRIIb 

The hyper-immune responsiveness toward the organismal elements (pneumococcal antigens and LPS) [38,43] and the non-infection related components (uremic toxin and drugs) [47,60] in FcgRIIb^−/−^ mice, due to the absence of the inhibitory signaling [29,43], are previously mentioned. Here, a single stimulation of Ahr activation alone did not induce inflammatory responses, in either FcgRIIb^−/−^ or WT macrophages, but LPS alone increased *Ahr* expression in macrophages from both mouse strains. Similarly, LPS pre-treatment before 1,4-CQ (LPS/1,4-CQ) predominantly enhanced the responses (cytokines and other pro-inflammatory parameters) against Ahr activator which were more prominent in FcgRIIb^−/−^ macrophages than the WT cells. Hence, the hyper-responsiveness of LPS/1,4-CQ stimulated FcgRIIb^−/−^ macrophages, at least in part, was a result of (i) LPS-enhanced *Ahr* expression for the subsequent 1,4-CQ stimulation in LPS-pretreatment protocol and (ii) amplification of activating-*FcgRs* without the inhibitory-*FcgRIIb*. In LPS/1,4-CQ, the LPS pre-stimulation upregulated *Ahr* (a receptor for 1,4-CQ) which might enhance the responses to the subsequent 1,4-CQ stimulation. Meanwhile, in 1,4-CQ/LPS, the 1,4-CQ pre-stimulation did not upregulate *TLR-4* (a receptor for LPS), and the responses to the subsequent LPS stimulation was not accelerated. Indeed, LPS enhances responses of TLR-4 [74] and activates both FcgRs [67,75] and Ahr [68], and the responses to LPS are more prominent in FcgRIIb^−/−^ macrophages than WT cells [66,68,76]. Additionally, a synergistic effect from TLR-4/FcgRs crosstalk is possible [66], and the blockage of the common downstream signaling from TLR-4 and activating FcgRs attenuates the responses of FcgRIIb^−/−^ macrophages [52,77]. Furthermore, LPS-induced cell apoptosis [53,54,55] was also detectable in LPS/1,4-CQ but not in N/1,4-CQ stimulation. These data imply a possible correlation between LPS, FcgRs, and Ahr for enhancing inflammation and cell apoptosis. Here, we demonstrated that LPS activation followed by an Ahr agonist (a representative environmental stimulator) could induce late apoptosis in FcgRIIb^−/−^ macrophages, but not in WT macrophages, which might accelerate activity of the lupus-like condition in FcgRIIb^−/−^ mice through apoptosis-induced lupus exacerbation [78]. The working hypothesis is demonstrated in Figure 9. Because both inflammatory cytokines and cell apoptosis are lupus exacerbation factors, the environmental activation (Ahr activation) might exacerbate lupus activity.

### 3.3. The Enhanced Activity of Lupus-Like Condition through the Aryl Hydrocarbon Receptor Activation, a Possible Impact of the Environmental Toxins in Lupus

An Ahr activator was administered in FcgRIIb^−/−^ and WT mice in a short- and long-term protocol. In a short-term administration, Ahr activation alone did not induce inflammatory cytokines similar to the *in vitro* experiments. Meanwhile, LPS pre-treatment before Ahr activation (LPS/1,4-CQ) induced the higher cytokines predominantly in FcgRIIb^−/−^ when compared with WT mice. In LPS/1,4-CQ activation, there was a higher inflammatory status in FcgRIIb^−/−^ mice over WT mice after the pre-treatment LPS injection, before the subsequent administration of Ahr agonist. Notably, LPS/1,4-CQ protocol on FcgRIIb^−/−^ mice was a stimulation upon an active inflammation (environmental toxins exposure during inflammation). These data also support the enhanced adverse effects of air pollutants (containing Ahr stimulator) in patients with ongoing active respiratory inflammation [71,72,73]. Although a short-term Ahr stimulation (without LPS) did not induce inflammation, a long-term Ahr stimulation increased serum cytokines in FcgRIIb^−/−^ mice but not in WT mice. Additionally, a long-term Ahr stimulation induced the lupus-like disease activity as indicated by anti-dsDNA, kidney immunoglobulin deposition, glomerular mesangial expansion, and cytokines in kidney tissue. Despite a less inflammatory effect of the short-term Ahr stimulation (without LPS), the accumulated injury from the long-term Ahr stimulation (chronic inflammation and atherosclerosis) in mice are demonstrated [80,81]. Because of the limited cell life-span, the *in vitro* demonstration of the accumulative injury is technically difficult and the *in vivo* experiments are necessary. Likewise, chronic exposure of PM2.5 (the particles containing Ahr agonists), induces oxidative stress, genotoxicity [82,83], and dysbiosis of gut microbiome (a possible mechanism of intestinal inflammation and gut permeability defect) [84,85]. Although endotoxemia from the long-term 1,4-CQ administration might be due to dysbiosis induction, the microbiome analysis was not performed here.

Nevertheless, the systemic inflammation in FcgRIIb^−/−^ mice after long-term 1,4-CQ stimulation was severe enough to exacerbate the lupus-like condition as indicated by increased production of auto-antibody and immune complex deposition-induced gut leakage [34,47]. Indeed, gut-leakage-induced endotoxemia could accelerate the severity of inflammation [51], in part, due to the potent inflammatory property of pathogen molecules [86]. Furthermore, serum LPS in FcgRIIb^−/−^ mice with long-term 1,4-CQ stimulation might be high enough to activate macrophage pro-inflammation that worsen lupus disease progression. Meanwhile, endotoxemia was not detectable in WT mice with long-term 1,4-CQ stimulation possibly due to the presence of the inhibitory FcgRIIb in WT mice. These data indicate (i) the pro-inflammatory induction of the long-term Ahr stimulation [80,81], but not in the short-term activation, and (ii) the lupus exacerbation through environment-induced inflammation [49,50]. Perhaps, a previous report on anti-inflammatory property of Ahr activator [87] might be due to a shorter duration of the administration, lack of the pre-conditioning stimulation and the different Ahr stimulators. Notably, FcgRIIb is a receptor for Fc portion of immunoglobulin G (IgG) that is detectable in all immune cells (except for T cells and NK cells) [75]. FcgRIIb is also detectable in some non-immune cells, including endothelium and hepatocyte (but not myocyte and adipocyte) [88,89]. Hence, an impact of 1,4-CQ in FcgRIIb^−/−^ mice might be due to the defect of FcgRIIb in other cell types, further studies on mice with FcgRIIb deficiency only in macrophages (Cre-Lox recombination) or adoptive transfer experiments are needed. Additionally, 1,4-CQ or other environmental toxins might also activate NK cells, another important innate immune cell [90,91]; however, FcgRIIb was not expressed on NK cells [75]. Other models are needed to explore the effect of Ahr activation on NK cells. Furthermore, FcgRIIb^−/−^ mice are only a model with lupus-like condition that are different from patients. The studies on patients with FcgRIIb de-functioning might be more informative for the environmental effect against lupus. 

Although further research on this topic is necessary for a solid conclusion, a proof of concept in a possible adverse effect from environmental toxins, which might be more profound in patients with lupus than the healthy persons, is demonstrated. Hence, a proper protection against environmental toxins for patients with lupus should be considered. 

## 4. Materials and Methods

### 4.1. The In Vitro Macrophage Experiments

Macrophages from RAW246.7 cell line (ATCC^®^, Manassas, VA, USA) and bone marrow-derived macrophages (BMM) were used. There were four groups of the cell treatment protocols (i) a single stimulation by 1,4-Chrysenequinone (1,4-CQ), an Ahr activator, (ii) a single stimulation by lipopolysaccharide (LPS), a positive control stimulator, and (iii) a sequential stimulation began with 1,4-CQ followed by LPS, a representative of toxin exposure before an inflammatory activation and iv) a sequential stimulation started with LPS followed by 1,4-CQ, a representative condition of inflammatory activation before the toxin exposure (Figure 1A). Indeed, the downstream of Ahr activation is partly associated with the signaling of LPS stimulation [92,93]. As such, LPS (*Escherichia coli* 026: B6; Sigma-Aldrich, St. Louis, MO, USA) at 100 ng/mL [34,47] and 1,4-CQ (Sigma-Aldrich) at 100 nM/mL following previous publications [1] were used. Because of the total 54 h incubation time of the sequential stimulation protocol (Figure 1A), the single-stimulation protocols were started with Dulbecco’s Modified Eagle Medium (DMEM) control for matching the condition (Figure 1A). As such, the single-stimulation protocol DMEM was initially used followed by 1,4-CQ or LPS for N/1,4-CQ and N/LPS group, respectively (Figure 1A). In LPS/1,4-CQ, LPS was incubated for 24 h before the further incubation by 6 h of DMEM, washed, and stimulated by 1,4-CQ before the sample collection (Figure 1A). In 1,4-CQ/LPS, the incubation was started by 1,4-CQ and followed by LPS (Figure 1A). In RAW264.7 experiments, the cells at 5 × 10^5^ cells/well were used. Meanwhile, BMM followed an established method was conducted [44,45]. Briefly, mouse bone marrow was flushed from the femurs and tibias of 8 week old mice, plated in petri dishes in DMEM-high glucose with 10% (*v*/*v*) fetal bovine serum (FBS), 1% (*v*/*v*) HEPES, 1% (*v*/*v*) sodium pyruvate and 1.3% (*v*/*v*) Pen-Strep, added macrophage colony-stimulating factor (M-CSF) (Sigma-Aldrich) at the 4th day and removed from petri dishes at the 7th day. Then, BMM at 1 × 10^6^ cells/well in 6-well plates were stimulated under 5% CO_2_ at 37 °C. At the end of the experiments, cells were harvested by cold phosphate buffer solution (PBS) before centrifugation at 800× *g*, 4 °C for 5 min and were stored at −80 °C until used.

Supernatant cytokines were measured by ELISA (PeproTech, Oldwick, NJ, USA) and expression of several genes were identified using real time polymerase chain reaction (PCR). As such, total RNA and the reverse transcription were prepared with an RNA-easy mini kit (Qiagen, Hilden, Germany) and a reverse-transcription assay (Applied Biosystems, Warrington, UK), respectively, using an Applied Biosystems QuantStudio 6 Flex Real-Time PCR System with SYBR^®^ Green PCR Master Mix (Applied Biosystems). The results were demonstrated in terms of relative quantitation of the comparative threshold (delta-delta Ct) method (2^−ΔΔCt^) as normalized by β-actin (an endogenous housekeeping gene). The list of primers is shown in Table 1. Moreover, BMM suspended in PBS at a concentration of 1 × 10^6^ cells/mL were stained for macrophage polarization by fluorescein isothiocyanate (FITC)-labeled CD206 (1 µL/well) and allophycocyanin (APC)-labeled CD86 antibodies (1 µL/well) (BD Biosciences, San Jose, CA, USA) for the M2 and M1 macrophage polarization, respectively, and stained for apoptosis/necrosis by annexin V-FITC and propidium iodide (PI) (5 µL/well) (BD Biosciences). Then, the samples were washed with FACS flow buffer, PBS supplemented with 1% (*v*/*v*) FBS and 0.05% NaN3, and processed in a BD LSR II Flow Cytometry (BD Biosciences) using the FloJo software (Tree Star Inc., Ashland, OR, USA).

### 4.2. Animal Model and Sample Collection

The animal study protocol was approved by the Institutional Animal Care and Use Committee of the Faculty of Medicine, Chulalongkorn University, Bangkok, Thailand, following the National Institutes of Health (NIH), USA. FcgRIIb-deficient (FcgRIIb^−/−^) mice on a C57BL/6 background were provided by Dr. Silvia Bolland (NIAID, NIH, Maryland, USA), and wild-type (WT) mice were purchased from Nomura Siam International (Pathumwan, Bangkok, Thailand). Female 8 week old FcgRIIb^−/−^ mice, asymptomatic lupus-prone mice [21,22,23,33,34], and the age-matched WT mice were used to explore the impact of Ahr activation in the administration in short term and long term. The short-term 1,4-CQ administration was designed for the comparison between 1,4-CQ alone versus 1,4-CQ after inflammatory stimulation (a representative for the environmental toxin exposure after another inflammatory stimulation). Here, LPS (*Escherichia coli* 026: B6) (Sigma-Aldrich) at 4 mg/kg was intraperitoneally (ip) administered [34,47] prior to an ip administration of 1,4-CQ (Sigma-Aldrich) at 1 mM/kg [1] at 24 h later (LPS/1,4-CQ). Then, the mice were sacrificed at 24 h after 1,4-CQ injection (Figure 6 schema). Due to the 48 h duration of LPS/1,4-CQ model (Figure 6 schema), other groups were following this pattern, including the single LPS stimulation (LPS/PBS, starting with LPS followed by PBS at 24 h later), the single 1,4-CQ (PBS/1,4-CQ, beginning with PBS followed by 1,4-CQ at 24 h later) and the PBS control (PBS/PBS, PBS was injected in both time-points). Notably, 1,4-CQ followed by PBS was not performed because 1,4-CQ effect was non-detectable at 48 h post-1,4-CQ injection (data not shown). Then, blood samples were collected through the tail-vein nicking before the second injection (0 h) and at 2, 6 and 24 h after the latter injection (Figure 6 schema) for the determination of serum cytokines by ELISA (PeproTech). For a short-term Ahr stimulation, total 50 mice were randomized into 5–7 mice per group.

For the long term 1,4-CQ stimulation, once daily ip administration of 1,4-CQ at 1 mM/kg or PBS control, following a publication [1], was performed for 8 weeks with blood collection via tail vein at several time-points and through cardiac puncture under isoflurane anesthesia at sacrifice (Figure 7 schema). A total of 36 mice were randomized into nine mice per group for WT (PBS-WT and 1,4-CQ WT) and seven mice per group for FcgRIIb^−/−^ mice (PBS-KO and 1,4-CQ-KO). The spot urine collection was performed by placing a mouse in a metabolic cage (Hatteras Instruments, NC, USA) for a while (in each time-points) and at 3 h before sacrifice. Lupus characteristics were determined by serum anti-dsDNA, serum creatinine, and proteinuria. Anti-dsDNA and serum creatinine were measured by the coated Calf-DNA (Invitrogen, Carlsbad, CA, USA) [94] and QuantiChrom Creatinine Assay (DICT-500) (BioAssay, Hayward, CA, USA), respectively. Proteinuria was calculated by spot urine protein creatinine index (UPCI) with an equation; UPCI = urine protein (µg/mg) urine creatinine (mg/dL). Urine protein and creatinine were measured by Bradford Bio-Rad Protein Assay (Bio-Rad, Hercules, CA, USA) and QuantiChrom Creatinine-Assay (DICT-500) (BioAssay), respectively. At sacrifice, kidneys were snap frozen and kept in −80 °C for tissue cytokine analysis, put in Cryogel (Leica Biosystems, Richmond, IL, USA) for fluorescent imaging and fixed in 10% formalin for histology. For kidney cytokines, the kidneys were washed several times in PBS, weighed, homogenized, and centrifuged for the determination of cytokines in the tissue (PeproTech). 

### 4.3. Gut Permeability Determination

Fluorescein isothiocyanate (FITC)-dextran, a gut non-absorbable molecule, was orally administered to determine gut permeability as previously published [65]. In brief, FITC-dextran (molecular weight 4.4 kDa; Sigma-Aldrich) was administered at 25 mg/mL in 0.25 mL PBS at 3 h before blood collection. Serum FITC-dextran was measured by fluorospectrometry (microplate reader; Thermo Scientific, Wilmington, DE, USA). In addition, serum endotoxin (LPS) was measured as another gut-leakage parameter using the Limulus Amebocyte lysate test (Associates of Cape Cod, East Falmouth, MA, USA) and values of LPS < 0.01 EU/mL were recorded as 0 due to the limitation of the standard curve.

### 4.4. Histology and Immunofluorescent Imaging

The kidney samples were prepared into 4 µm thick paraffin-embedded sections before staining with Hematoxylin and Eosin (H&E) for semi-quantitative evaluation through the percentage of glomeruli with expansion as determined by glomerular size and the prominence of mesangial area at 400× magnification. All glomeruli in the slide were examined and presented as the percentage of glomerular expansion. Notably, there was no severe glomerular changes and non-tubular injury in all mice (data not shown). In parallel, immunoglobulin deposition in kidneys was visualized by immunofluorescence stained with goat anti-mouse IgG and DAPI (4′,6-diamidino-2-phenylindole), a blue-fluorescent DNA staining color (Alexa Fluor 488; Abcam, Cambridge, MA, USA). Then, the fluorescent intensity was analyzed by ZEISS LSM 800 (Carl Zeiss, Germany). All stained glomeruli in the section were scored. The antibody deposition in FcgRIIb^−/−^ mice with high anti-dsDNA indicates immune complex deposition [51].

### 4.5. Statistical Analysis

Statistical differences among groups were examined using the unpaired Student’s t-test or one-way analysis of variance (ANOVA) with Tukey’s comparison test for the analysis of experiments with two groups or more than two groups, respectively, and are presented as the mean ± standard error (SE). The time-point experiments were analyzed by the repeated measures ANOVA. All statistical analyses were performed with SPSS 11.5 software (SPSS, IL, USA) and Graph Pad Prism version 7.0 software (La Jolla, CA, USA). A *p*-value of < 0.05 was considered statistically significant.

## 5. Conclusions

In conclusion, our data supported the prominent hyperinflammatory responses to Ahr stimulation in FcgRIIb^−/−^ mice (a lupus-like model), over WT mice, especially with the LPS pre-treatment. Because Ahr is a major environmental sensor that activates immune responses, the prominent inflammatory effect of Ahr activator in FcgRIIb^−/−^ macrophages and in mice when compared with WT groups indicated a possible more severe adverse effect of environmental toxins in patients with lupus. More studies on the impact of environmental toxins on patients with lupus would be interesting. 

## Figures and Tables

**Figure 1 ijms-22-04199-f001:**
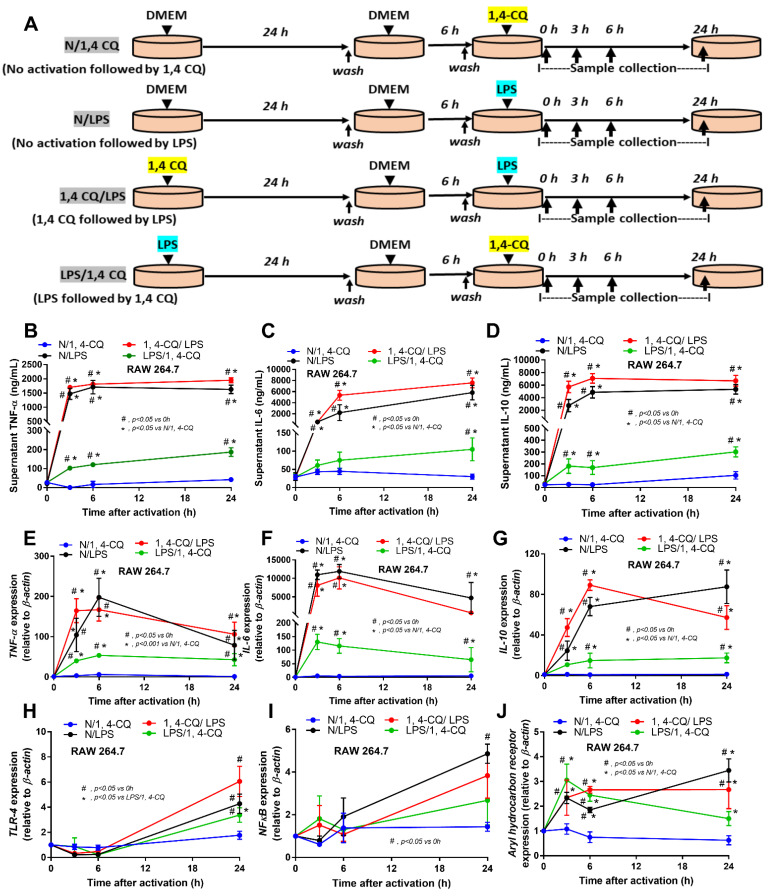
Schema of the in vitro experiments (**A**) (details in Materials and Methods) and the characteristic responses of RAW246.7, a macrophage cell line, against a single activation by an aryl hydrocarbon receptor activator; 1,4-chrysenequinone (N/1,4-CQ), or lipopolysaccharide (N/LPS) (a positive-control stimulator) or the activation with the pre-treatment protocols (1,4-CQ/LPS and LPS/1,4-CQ) as determined by supernatant cytokines (**B**–**D**) and gene expression of inflammatory cytokines (**E**–**G**) and the downstream signaling molecules (*TLR-4*, *NF-κB* and *aryl hydrocarbon receptor*) (**H**–**J**) are demonstrated. DMEM, Dulbecco’s Modified Eagle Medium (culture media of control group). All experiments were independently performed in triplicate.

**Figure 2 ijms-22-04199-f002:**
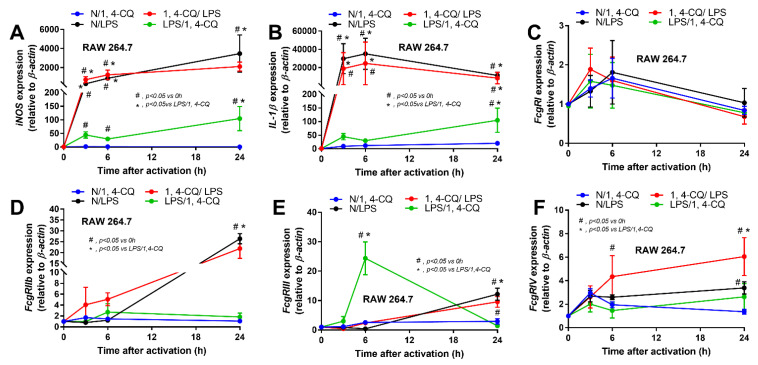
The characteristic responses of RAW246.7, a macrophage cell line, against a single activation by an aryl hydrocarbon receptor activator; 1,4-chrysenequinone (N/1,4-CQ), or a positive-control stimulator, lipopolysaccharide (N/LPS), and the activation with the pre-treatment protocols (1,4-CQ/LPS and LPS/1,4-CQ) as demonstrated by gene expression of M1 macrophage polarization (*iNOS* and *IL-1β*) (**A**,**B**) and Fc gamma receptors (*FcgRs*) (**C**–**F**) are demonstrated. DMEM, Dulbecco’s Modified Eagle Medium (culture media of control group). All experiments were independently performed in triplicate.

**Figure 3 ijms-22-04199-f003:**
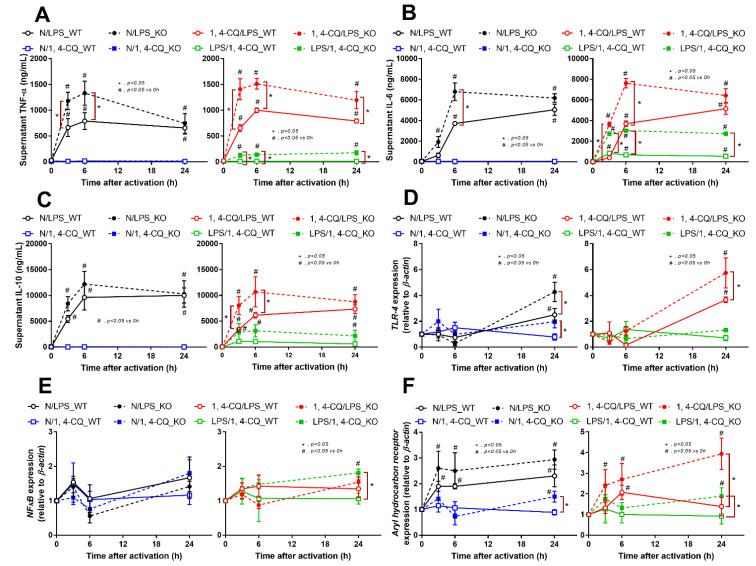
Characteristics of the responses in FcgRIIb^−/−^ lupus macrophages (KO) and wild-type cells (WT) after a single activation by an aryl hydrocarbon receptor activator, 1,4-chrysenequinone (N/1,4-CQ), or lipopolysaccharide (N/LPS) (a positive-control stimulator) (left side of each graph) and the activation after the pre-treatment protocols (1,4-CQ/LPS and LPS/1,4-CQ) (right side of each graph) as determined by supernatant cytokines (**A**–**C**) and gene expression of the downstream signaling molecules (*TLR-4*, *NF-κB* and *aryl hydrocarbon receptor*) (**D**–**F**) are demonstrated. All experiments were independently performed in triplicate.

**Figure 4 ijms-22-04199-f004:**
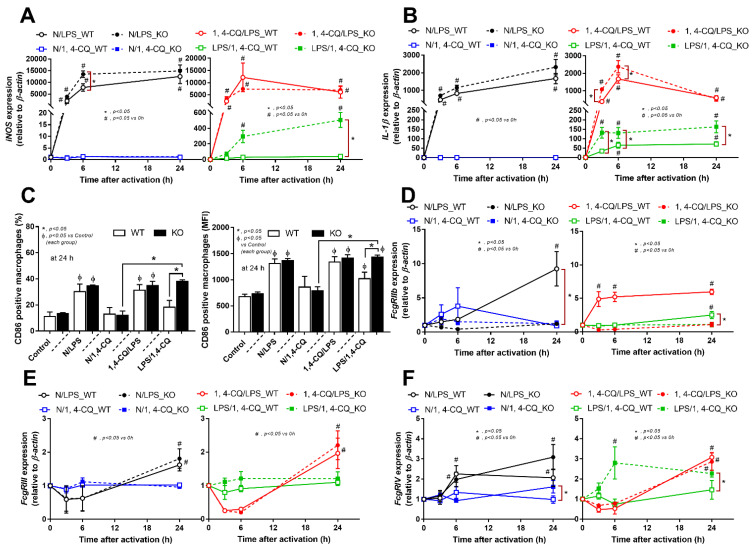
Characteristics of the responses in FcgRIIb^−/−^ macrophages (KO) and wild-type cells (WT) after a single activation by an aryl hydrocarbon receptor activator, 1,4-chrysenequinone (N/1,4-CQ), or a positive-control stimulator, lipopolysaccharide (N/LPS) and the activation after the pre-treatment protocols (1,4-CQ/LPS and LPS/1,4-CQ) as determined by the expression of macrophage polarization genes for M1 polarization (*iNOS* and *IL-1β*), CD-86 (M1 macrophage polarization marker) at 24 h post-stimulation (flow cytometry analysis) (**A**–**C**) and expression of Fc gamma receptors (*FcgRs*) (**D**–**F**) are demonstrated. All experiments were independently performed in triplicate.

**Figure 5 ijms-22-04199-f005:**
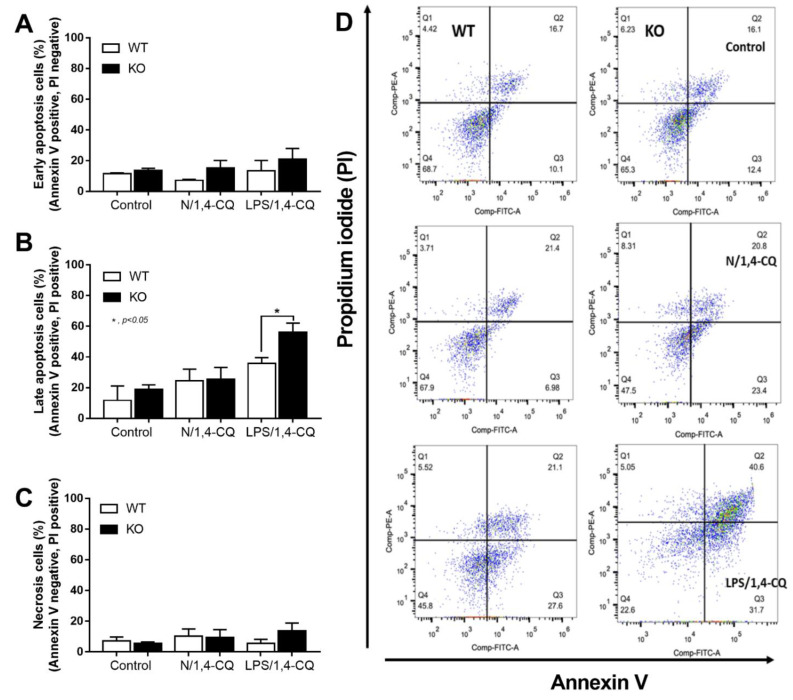
The quantitative flow-cytometric analysis of FcgRIIb^−/−^ macrophages (KO) and wild-type cells (WT) after control media incubation (control), a single activation by an aryl hydrocarbon receptor activator, 1,4-chrysenequinone (N/1,4-CQ), or the 1,4-CQ activation after LPS pre-treatment (LPS/1,4-CQ) as stained by Annexin V and propidium iodide (PI) for the determination of early apoptosis cells (Annexin V positive, PI negative), late apoptosis cells (Annexin V positive, PI positive) and necrotic cells (Annexin V negative, PI positive) (**A**–**C**) with the representative flow-cytometric patterns (**D**) are demonstrated. All experiments were independently performed in triplicate.

**Figure 6 ijms-22-04199-f006:**
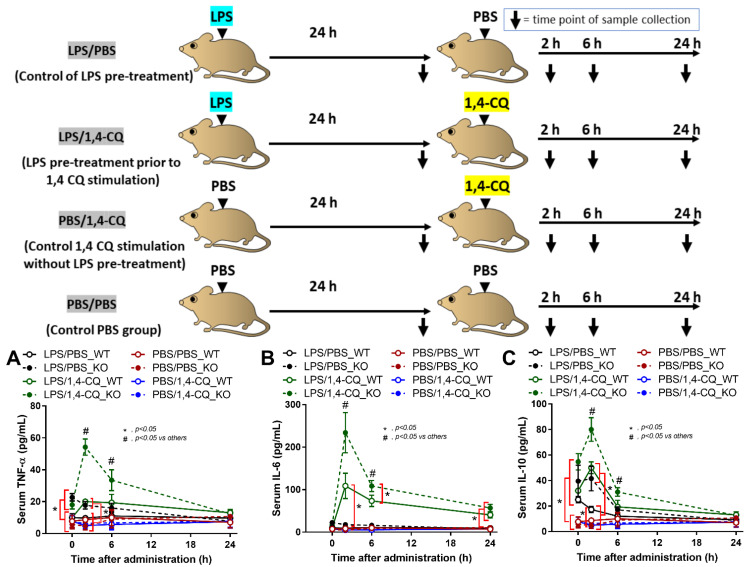
Schema of the short-term experiments (upper part of figure) and the characteristics of FcgRIIb^−/−^ mice (KO) and wild-type mice (WT) mice after the pre-treatment with ip administration of lipopolysaccharide (LPS) or control phosphate buffer solution (PBS) at 24 h before ip injection with an aryl hydrocarbon receptor activator, 1,4-chrysenequinone (1,4-CQ), or PBS as indicated by serum cytokines (TNF-α, IL-6 and IL-10) (**A**–**C**) (n = 5–7/time-point) are demonstrated. Abbreviation of the protocols are PBS/PBS (control), PBS injection prior to PBS; PBS/1,4-CQ (single 1,4-CQ administration), PBS injection prior to 1,4-CQ; LPS/1,4-CQ (LPS pre-treatment before 1,4-CQ), LPS injection prior to 1,4-CQ; LPS/PBS (control of LPS pre-treatment), LPS injection prior to PBS.

**Figure 7 ijms-22-04199-f007:**
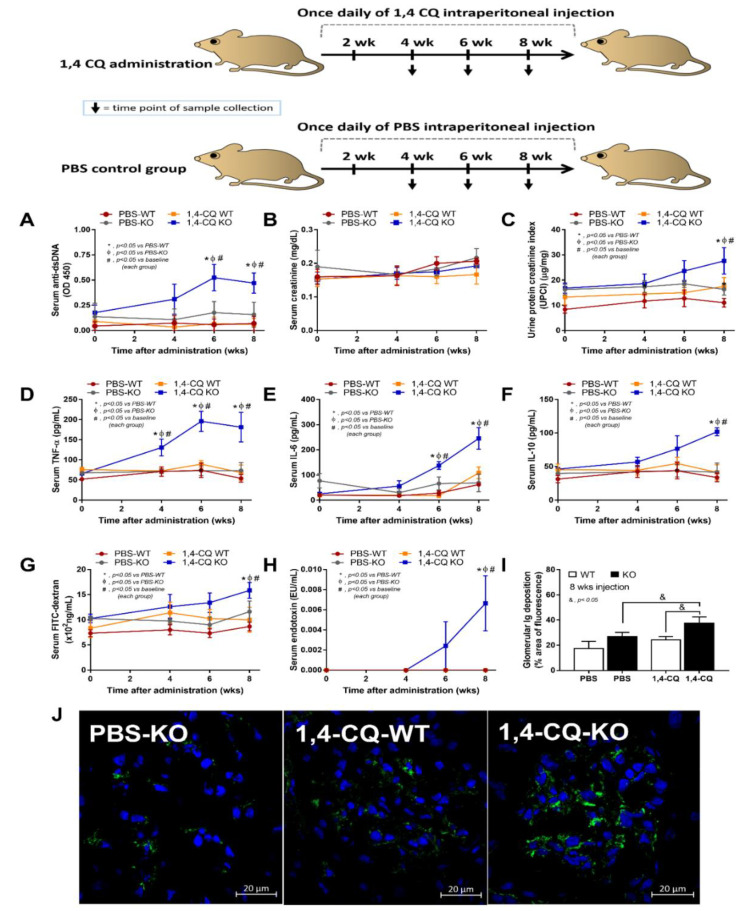
Schema of the long-term daily ip administration for 8 weeks with aryl hydrocarbon receptor activator, 1,4-chrysenequinone (1,4-CQ) or phosphate buffer solution (PBS) control (upper part of the figure) and the characteristics of FcgRIIb^−/−^ mice (KO) and wild-type mice (WT) as determined by lupus characteristics (anti-dsDNA, serum creatinine, and urine protein creatinine index) (**A**–**C**), serum cytokines (**D**–**F**), gut leakage (FITC-dextran assay and endotoxemia) (**G**,**H**), glomerular immunoglobulin deposition score (**I**), and the representative immunofluorescent glomerular pictures (**J**) are demonstrated (n = 7–9/time point or group). The original magnification of the glomeruli is 200x; green and blue color demonstrate mouse IgG and glomerular nuclei, respectively. The picture of PBS-administered wild-type control mice (PBS-WT) is not shown due to the similarity to PBS-KO group.

**Figure 8 ijms-22-04199-f008:**
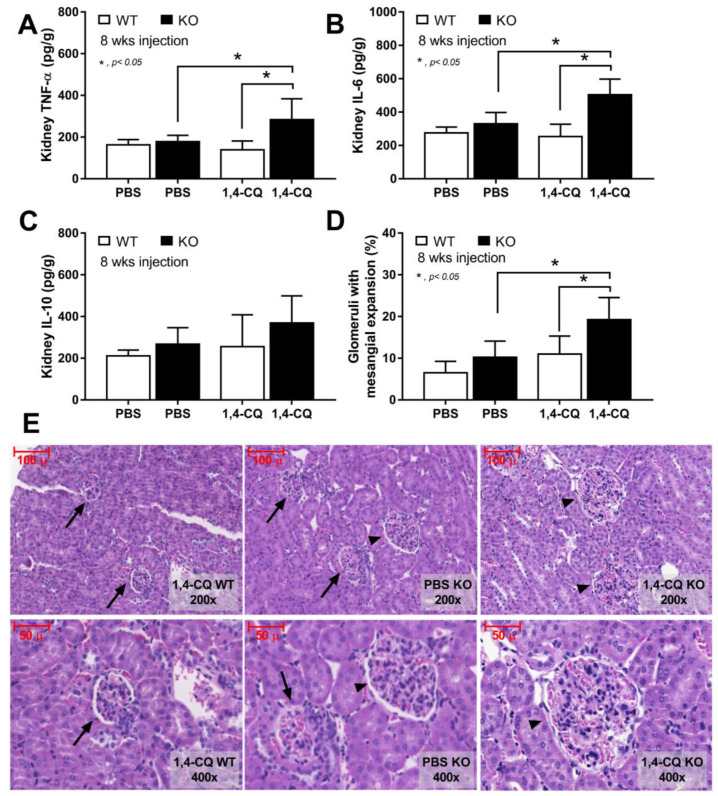
The characteristics of FcgRIIb^−/−^ mice (KO) and wild-type mice (WT) after daily 8 week administration of aryl hydrocarbon receptor activator, 1,4-chrysenequinone (1,4-CQ) or phosphate buffer solution (PBS) control as determined by cytokines in renal tissue (**A**–**C**) and percentage of glomeruli with glomerular mesangial expansion with the representative H&E-stained histological pictures (original magnification are at 200× and 400×) (**D**,**E**) are demonstrated (n = 7–9/group). The pictures of PBS administered WT mice (PBS-WT) are not shown due to the similarity to 1,4-CQ WT group. Arrow head, glomeruli with mesangial expansion; arrow, normal glomeruli.

**Figure 9 ijms-22-04199-f009:**
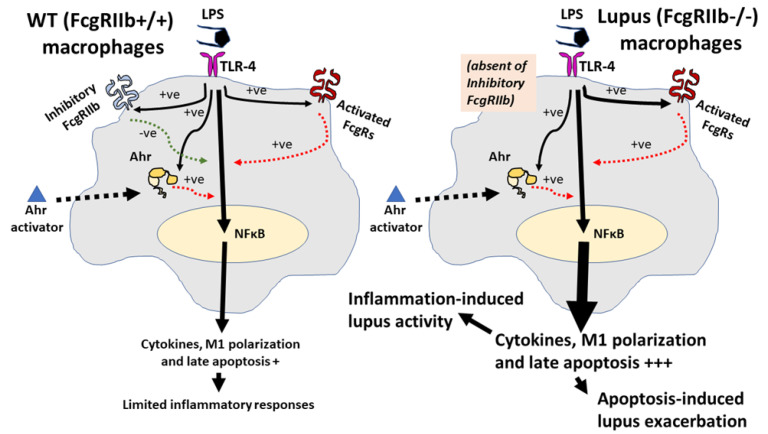
The proposed hypothesis demonstrates a difference between FcgRIIb^−/−^ macrophages and wild-type (WT) macrophages. In WT, the pre-treatment by lipopolysaccharide (LPS) activates TLR-4, aryl hydrocarbon receptor (Ahr) and Fc gamma receptors (FcgRs), including the activating-FcgRs and an inhibitory-FcgRIIb [66,68,76,79], partly through NF-κB signaling. Then, the enhanced Ahr by LPS accelerates responses against a subsequent Ahr stimulation. Without the inhibitory signaling, inflammatory reaction is increased and inflammation-induced apoptosis is more prominent in FcgRIIb^−/−^ cells than WT [53,54,55]. Both inflammation and apoptosis possibly exacerbate activity of the lupus-like condition [78]: Solid and dot line in black demonstrate direction of the activation by LPS and Ahr activator, respectively. Red- and green-colored dotted lines are an activating and inhibitory signaling, respectively. The symbol +++ and + are the estimated higher and lower intensity of the reaction, respectively.

**Table 1 ijms-22-04199-t001:** List of primers used in the study.

Name	Forward Primer	Reverse Primer
Arginase-1 (***Arg-1***)	5′-CTTGGCTTGCTTCGGAACTC-3′	5′-GGAGAAGGCGTTTGCTTAGTT-3′
Aryl hydrocarbon receptor (***Ahr***)	5′-GACCACTTAGAGCACCACTA-3′	5′-AGAACTTCAATCAGACATACACAA-3′
Fc gamma receptor I (***FcgRI***)	5′-CACAAATGCCCTTAGACCAC-3′	5′-ACCCTAGAGTTCCAGGGATG-3′
Fc gamma receptor IIb (***FcgRIIb***)	5′-TTCTCAAGCATCCCGAAGCC-3′	5′-TTCCCAATGCCAAGGGAGAC-3′
Fc gamma receptor III (***FcgRIII***)	5′-AGGGCCTCCATCTGGACTG-3′	5′-GTGGTTCTGGTAATCATGCTCTG-3′
Fc gamma receptor IV (***FcgRIV***)	5′-AACGGCAAAGGCAAGAAGTA-3′	5′-CCGCACAGAGAAATACAGCA-3′
Inducible nitric oxide synthase (***iNOS***)	5′-ACCCACATCTGGCAGAATGAG-3′	5′-AGCCATGACCTTTCGCATTAG-3′
Interleukin-1β (***IL-1β***)	5′-GAAATGCCACCTTTTGACAGTG-3′	5′-TGGATGCTCTCATCAGGACAG-3′
Interleukin-6 (***IL-6***)	5′-TACCACTTCACAAGTCGGAGGC-3′	5′-CTGCAAGTGCATCATCGTTGTTC-3′
Interleukin-10 (***IL-10***)	5′-GCTCTTACTGACTGGCATGAG-3′	5′-CGCAGCTCTAGGAGCATGTG-3′
Nuclear factor-κB (***NF-κB RelA***)	5′-CTTCCTCAGCCATGGTACCTCT-3′	5′-CAAGTCTTCATCAGCATCAAACTG-3′
Resistin-like molecule-α (***FIZZ-1***)	5′-GCCAGGTCCTGGAACCTTTC-3′	5′-GGAGCAGGGAGATGCAGATGA-3′
Toll like receptor 4 (***TLR-4***)	5′-GGCAGCAGGTGGAATTGTAT-3′	5′-AGGCCCCAGAGTTTTGTTCT-3′
Transforming Growth Factor-β (***TGF-β***)	5′-CAGAGCTGCGCTTGCAGAG-3′	5′-GTCAGCAGCCGGTTACCAAG-3′
Tumor necrosis factor α (***TNF-α***)	5′-CCTCACACTCAGATCATCTTCTC-3′	5′-AGATCCATGCCGTTGGCCAG-3′
β-actin	5′-CGGTTCCGATGCCCTGAGGCTCTT-3′	5′-CGTCACACTTCATGATGGAATTGA-3′

## Data Availability

The data presented in this study are available on request from the corresponding author.

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
