# Peer review of "Lipopolysaccharide-Enhanced Responses against Aryl Hydrocarbon Receptor in FcgRIIb-Deficient Macrophages, a Profound Impact of an Environmental Toxin on a Lupus-Like Mouse Model"

_ijms, 2021, doi:10.3390/ijms22084199_

Round 1

Reviewer 1 Report

The submitted original research by Udompornpitak et al. entitled Lipopolysaccharide enhanced responses against aryl hydrocarbon receptor in FcgRIIb deficient macrophages, a profound impact of an environmental toxin on lupus, raises an interesting issue of the role of environmental toxins in the pathogenesis of lupus. Even though the topic itself is interesting, the present form of the manuscript prevents publishing, especially in the International Journal of Molecular Sciences.

The major flaw is the form of the manuscript and the language use, as it makes this paper completely incomprehensible. It requires thorough editing and some simplifications throughout the text.

Other important issues consider the methodology of the study. It is ones again, difficult to understand and follow. I cannot understand the experimental schemes. Could the authors be more precise in the description? I do not follow, even if I try to use the experiment’s graphic presentation – it should be enhanced with details. Additionally, the in vivo experiment should also be presented as an experimental scheme.

Moreover, how many technical and biological replicates were done in the study? This question applies to both in vitro and in vivo studies. What was the sample size that allowed the authors to use the graphical presentation of data restricted to the parametrical test?

I do not know whether these flaws are so detectable due to inappropriate experimental design or language problems. I hope the authors can introduce changes making the manuscript more digestible for the reader.

Author Response

Reviewer 1

The submitted original research by Udompornpitak et al. entitled Lipopolysaccharide enhanced responses against aryl hydrocarbon receptor in FcgRIIb deficient macrophages, a profound impact of an environmental toxin on lupus, raises an interesting issue of the role of environmental toxins in the pathogenesis of lupus. Even though the topic itself is interesting, the present form of the manuscript prevents publishing, especially in the International Journal of Molecular Sciences.

  1. The major flaw is the form of the manuscript and the language use, as it makes this paper completely incomprehensible. It requires thorough editing and some simplifications throughout the text.

ANS. We extremely apologize for the unclear presentation and our limitation of the English language. We thank the reviewer for an opportunity to edit the manuscript to reduce the language barrier in the scientific community. We rewrite all parts of manuscript, draw all schemas of the experiments and send to the English editing service of the university before the submission of the revision. We are willing to seek more help in the English editing service if quality of the English language is still needed for an improvement.

  1. Other important issues consider the methodology of the study. It is ones again, difficult to understand and follow. I cannot understand the experimental schemes. Could the authors be more precise in the description? I do not follow, even if I try to use the experiment’s graphic presentation – it should be enhanced with details. Additionally, the in vivo experiment should also be presented as an experimental scheme.

ANS. We extremely apologize for the unclear presentation. We put the schemas in all experiments and mention more details in the method section.

  1. Moreover, how many technical and biological replicates were done in the study? This question applies to both in vitro and in vivo studies. What was the sample size that allowed the authors to use the graphical presentation of data restricted to the parametrical test?

ANS. We apologize for the unclear presentation and mention the replicates and number of animals in all figure legends. Because all mice were inbred and the total number of mice in each experiment were more than 30 mice, there was a normal distribution in all set of the data. 

  1. I do not know whether these flaws are so detectable due to inappropriate experimental design or language problems. I hope the authors can introduce changes making the manuscript more digestible for the reader.

ANS. We extremely apologize for the unclear presentation and our limitation of the English language. Hopefully, the new version of our manuscript could reach the minimum requirement for the publication in the journal. Our data will lead to the more concerns toward the environmental toxins for the special group of patients.

Reviewer 2 Report

This manuscript describes the effect of LPS against aryl hydrocarbon receptor in FcgR2b deficient macrophages using cell lines (RAW 264.7 cells) and mouse model (FcgR2b knock out mice and their macrophages). The result is of interesting to readers, however, several points should be more clearly discussed. Comments are listed below.  

Major 

  1. In vitro studies, LPS stimulation is very affected in production of cytokines (such as TNF-alpha, IL-6) in RAW264.7 cells and macrophages (WT and KO). But LPS pretreatment seems not to be so strong increase of cytokines. And some cytokines may not show significant changes in mRNA expression in LPS/1,4 CQ stimulation (TNF-alpha and IL-6 in Figure 3B, D). How LPS worked in these macrophages?         
  2. There are so many graphs, and that hampers what the authors want to demonstrate. Please reconsider the graphs including dots and lines (in the point of visibility). Some information can be moved to supplementary file.
  3. What is ve (an abbreviation)?
  4. Figure 7, we understand the increase of glomerular Ig deposition in 1,4 CQ KO mice. How about the glomerular inflammation of lupus (by scoring, or other staining)? It would be more informative if the grade of inflammation is associated with Ig deposition.   

Author Response

Reviewer 2

This manuscript describes the effect of LPS against aryl hydrocarbon receptor in FcgR2b deficient macrophages using cell lines (RAW 264.7 cells) and mouse model (FcgR2b knock out mice and their macrophages). The result is of interesting to readers, however, several points should be more clearly discussed. Comments are listed below.  

Major 

  1. In vitro studies, LPS stimulation is very affected in production of cytokines (such as TNF-alpha, IL-6) in RAW264.7 cells and macrophages (WT and KO). But LPS pretreatment seems not to be so strong increase of cytokines. And some cytokines may not show significant changes in mRNA expression in LPS/1,4 CQ stimulation (TNF-alpha and IL-6 in Figure 3B, D). How LPS worked in these macrophages? 

ANS. We apologize for the unclear presentation. While LPS/1,4-CQ shows a less effect, N/LPS shows a potent effect. This is because of the difference between the time of the observation as we clarify on the new schema in fig 1. In the single LPS stimulation (N/LPS), the samples were evaluated at 24h post LPS-incubation. Meanwhile, the samples in the LPS/1,4 CQ were collected at 54 h post LPS stimulation (with the 6 h of wash-out period). Hence, there was very less effect of LPS in LPS/1,4 CQ. Of note, the main comparison is N/1,4-CQ vs LPS/1,4-CQ. Then, we explain this point more in the result section of the figure 1 for the better understanding as following “Notably, N/LPS was evaluated at 24 h post LPS, while, in LPS/1,4-CQ, samples were collected at 54 h post LPS with a wash-out procedure using 6 h DMEM (fig 1A) to set a baseline of all markers as indicated by the low supernatant cytokines at the 0 time-point in LPS/1,4-CQ group (fig 1B-D). The comparison between LPS/1,4-CQ and N/1,4-CQ represents the cells with pre-stimulation versus the regular cells, respectively.”.

  1. There are so many graphs, and that hampers what the authors want to demonstrate. Please reconsider the graphs including dots and lines (in the point of visibility). Some information can be moved to supplementary file.

ANS. We thank the reviewer for the comment. We reduce the complexity of the figures and move some information into the supplements.

  1. What is ve (an abbreviation)?

ANS. We apologize for the unclear presentation and use the full word (positive and negative).

  1. Figure 7, we understand the increase of glomerular Ig deposition in 1,4 CQ KO mice. How about the glomerular inflammation of lupus (by scoring, or other staining)? It would be more informative if the grade of inflammation is associated with Ig deposition. 

ANS: We thank the reviewer for the comment. Indeed, there was a little bit change on the percentage of glomerular expansion in the long-term 1,4-CQ administered FcgRIIb-/- mice. Then we add the data in the new figure 8 of the manuscript  

Reviewer 3 Report

The author found that the activation of aryl hydrocarbon receptor worsened lupus severity in FcgRIIb-/- mice through the enhanced inflammatory by LPS and then suggested patients with lupus might be more vulnerable to the environmental pollution.

The data is interesting, but the conclusion is overinterpreted if there is no more relevant data to support.

  1. How is the phenotype only related to macrophages? Why the impact of an environmental toxin on lupus is only from macrophages, but not from other innate immune cells such as NK cells?
  2. How about the function of FcgRIIb in non-hematologic cells? The author should perform BMT exp to transfer FcgRIIb-/- bone marrow into wt recipient mice and vice versa under the toxin condition and check serum endotoxin or FITC-dextran as shown in fig7. This exp is important in that it tells if the effect comes from hematopoietic cells at least although it is still not macrophage specific.
  3. How the FcgRIIb-/- macrophages that display drastic responses with LPS/14-CQ are relevant to human with lupus? Is there any evidence to demonstrate that in humans, lupus patents that are deficient in FcgRIIb have higher chance to get worse with or without environment influences? Macrophages with no FcgRIIb seems more responsive to LPS, but is this relevant to lupus patients under the clinical setting? If not, the author is studying the function of FcgRIIb or its suppressive signaling, not really lupus disease; after all, FcgRIIb intact macrophages lack phenotypes even after treating with LPS and 14-CQ.
  4. It seems to me that the impact of an environmental toxin in macrophages that are deficient in FcgRIIb is minimum if there is no LPS (fig4), suggesting that LPS is required. However, is there any evidence to show that lupus patents or FcgRIIb-/- mice have higher LPS concentrations in vivo and higher enough to cause macrophage activation to worsen disease progression?
  5. Is the concentration of 14-CQ used to treat mice physiologically relevant in fig7?
  6. Fig3 showed macrophages with/without FcgRIIb that are cultured with 14-CQ did not show much inflammation, but in fig7, why 14-CQ causes increased plasma endotoxin even without LPS? Does that mean the gut leakage phenotype is driven by FcgRIIb-/- in other cell types? This data is interesting, but does not seem to be aligned with previous figs to me.
  7. The statistics labeling needs to be greatly improved, so does the number of mice used and the biological repeats .

Author Response

Reviewer 3

The author found that the activation of aryl hydrocarbon receptor worsened lupus severity in FcgRIIb-/- mice through the enhanced inflammatory by LPS and then suggested patients with lupus might be more vulnerable to the environmental pollution.

The data is interesting, but the conclusion is overinterpreted if there is no more relevant data to support.

  1. How is the phenotype only related to macrophages? Why the impact of an environmental toxin on lupus is only from macrophages, but not from other innate immune cells such as NK cells?

ANS. We thank the reviewer for this important comment. Macrophages have been mentioned as a major cell responsible for environments-induce inflammation possibly due to the phagocytosis activity to engulf the toxins in any sizes and the ability to produce cytokines. Although there are data on the influence of environmental toxins toward NK cells, FcgRIIb is not presented on NK cell.

Then, we add some information; including i) add more information on the importance of macrophages in inflammation in the introduction as following “It is interesting to note that macrophages are major innate immune cells responsible for the pollutants-induced inflammation, possibly due to the professional phagocytosis activity and cytokine production [19-21]. Among phagocytic cells (macrophage, neutrophil and dendritic cell) [22, 23], macrophages initiate inflammation for the subsequent healing processes [24, 25] and are distributed throughout the body refer to as “sentinel immune cells” [26, 27]. Despite the various forms of Ahr agonists with sever-al routes of the contamination, these substances could induce macrophages in several parts of the body; including liver (Kupffer cells), brain (microglia), kidney (mesangial cells), lung (alveolar macrophages) and intestine (intestinal macrophages) [28].”

  1. ii) mention the possible importance of NK cells against the environmental toxins in the discussion as following “Additionally, 1,4-CQ or other environmental toxins might also activate NK cells, an-other important innate immune cell [90, 91]; however, FcgRIIb was not expressed on NK cells [74]. Other models are needed to explore the effect of Ahr activation on NK cells. “

  1. How about the function of FcgRIIb in non-hematologic cells? The author should perform BMT exp to transfer FcgRIIb-/- bone marrow into wt recipient mice and vice versa under the toxin condition and check serum endotoxin or FITC-dextran as shown in fig7. This exp is important in that it tells if the effect comes from hematopoietic cells at least although it is still not macrophage specific.

ANS. We thank the reviewer for the comment. Indeed, FcGRIIb in endothelium and hepatocyte might also important for the impact of Ahr activation. However, we have a limitation in our funding to perform BMT adoptive transfer. Hence, we put this note an a limitation of our manuscript at the discussion as following “FcgRIIb is detectable in some non-immune cells; including endothelium and hepatocyte (but not myocyte and adipocyte) [88, 89]. Hence, an impact of 1,4-CQ in FcgRIIb-/- mice might due to the FcgRIIb defect in other cell types, the further studies on mice with FcgRIIb deficiency only in macrophages (Cre-Lox recombination) or adoptive transfer experiments are needed.”

  1. How the FcgRIIb-/- macrophages that display drastic responses with LPS/14-CQ are relevant to human with lupus? Is there any evidence to demonstrate that in humans, lupus patients that are deficient in FcgRIIb have higher chance to get worse with or without environment influences? Macrophages with no FcgRIIb seems more responsive to LPS, but is this relevant to lupus patients under the clinical setting? If not, the author is studying the function of FcgRIIb or its suppressive signaling, not really lupus disease; after all, FcgRIIb intact macrophages lack phenotypes even after treating with LPS and 14-CQ.

ANS. We apologize for the unclear presentation. Our works are following a previous work that mentions the influence of macrophages with FcgRIIb defect by Willcocks et al. (Willcocks LC, A de-functioning polymorphism in FCGR2B is associated with protection against malaria but susceptibility to systemic lupus erythematosus. Proc Natl Acad Sci U S A. 2010 Apr 27;107(17):7881-5.). They demonstrate that the hyperfunction of macrophages with FcgRIIb defect kill malaria better but the hyper-immune responses increase an incidence of lupus. Furthermore, the impact of environmental factors to lupus exacerbation is well-known (Parks CG, Occupational exposures and risk of systemic lupus erythematosus. Autoimmunity. 2005, Parks CG, Understanding the role of environmental factors in the development of systemic lupus erythematosus. Best Pract Res Clin Rheumatol. 2017). They mention that silica, current cigarette smoking, air pollution, ultraviolet light, solvents, pesticides, and heavy metals, may increase SLE risk (incidence and exacerbation). However, there is very limited data on environmental impact on patients with the defects on a specific gene, including FcgRIIb, as multiple-genes are responsible for lupus and the genetic test has not regularly been done in the current clinical practice. Nevertheless, we totally agree that the study on FcgRIIb-/- is not a real lupus disease (it might be more suitable to use a term “lupus-like condition”).

Hence, we edit several parts as shown in the new version of the manuscript.

  1. i) we remind the reader that our study currently demonstrated the effect of inhibitory FcgRIIb, but not directly lupus, including
  2. Change several parts that use the term “lupus” into “lupus-like condition” (including at the title)
  3. We change all terms with “FcgRIIb-/- lupus” into “FcGRIIb-/-“.
  4. We put a remark on the lack of studies on patients with FcgRIIb defect at the limitation part of the discussion as following “Furthermore, FcgRIIb-/- mice are only a model with the lupus-like condition, the stud-ies on patients or macrophages on the patients with FcgRIIb de-functioning might be more informative for the conclusion on the environmental effect against lupus.”

iii) We also mention human data on macrophages in the introduction as following “In human, the FcgRIIb defects are associated with increased incidence of lupus but more effective in the protection against malaria and several infections than the control group  [31, 37-39], implying a possible hyper-activity of macrophages with FcgRIIb defects.”.  

  1. It seems to me that the impact of an environmental toxin in macrophages that are deficient in FcgRIIb is minimum if there is no LPS (fig4), suggesting that LPS is required. However, is there any evidence to show that lupus patients or FcgRIIb-/- mice have higher LPS concentrations in vivo and higher enough to cause macrophage activation to worsen disease progression?

ANS. We thank the reviewer for the comment. The endotoxemia in patients with active lupus is mentioned in “Shi L, The SLE transcriptome exhibits evidence of chronic endotoxin exposure and has widespread dysregulation of non-coding and coding RNAs. PLoS One 2014”. Following this article, we support gut translocation in patients with active lupus and in FcgRIIb-/- mice in “Issara-Amphorn J, The synergy of endotoxin and (1 → 3)-β-D-glucan, from gut Translocation, worsens sepsis severity in a lupus model of Fc gamma receptor IIb-deficient mice, J of Innate immunity, 2017”. Hence, we put a remark on this topic in the introduction as following “Endotoxemia in patients with active lupus, [46], at least in part, supports a possible importance of LPS and gut permeability defect in patients.”

  1. Is the concentration of 14-CQ used to treat mice physiologically relevant in fig7?

ANS. We apologize for the unclear presentation. The dose of chronic Ahr activation is following a study “Wu, D.; Activation of aryl hydrocarbon receptor induces vascular inflammation and promotes atherosclerosis in apolipoprotein E-/- mice. Arterioscler Thromb Vasc Biol 2011”. We then put a remark sentence in the method section and put this reference on the section.

  1. Fig3 showed macrophages with/without FcgRIIb that are cultured with 14-CQ did not show much inflammation, but in fig7, why 14-CQ causes increased plasma endotoxin even without LPS? Does that mean the gut leakage phenotype is driven by FcgRIIb-/- in other cell types? This data is interesting, but does not seem to be aligned with previous figs to me.

ANS. We thank the reviewer for the comment. We hypothesize that this is due to the difference between the single exposure and the chronic exposure of the substance. While the in vitro represents only the acute exposure effect, the effect of the long-term exposure (8 wks) could not be done in the in vitro experiments. Indeed, there are several publications demonstrate the pro-inflammatory effect of the Ahr activations. Hence, we put a note on this discrepancy in the discussion of the manuscript as following “Despite the less inflammatory effect of the activation by Ahr agonist alone (without LPS) in macrophages and in the short-term Ahr stimulation, the long-term administra-tion of Ahr agonist might induce accumulation of the responses which is a technically limitation for the experiments on macrophages due to the limited duration of cell via-bility. Similarly, chronic inflammation and atherosclerosis in mice after long-term Ahr stimulation are previously published [80, 81].”.

  1. The statistics labeling needs to be greatly improved, so does the number of mice used and the biological repeats.

ANS. We thank the reviewer for the comment. We simplify the labeling and demonstrate number of mice and the biological repeats in the figure legends.

Round 2

Reviewer 1 Report

The Authors put much effort into the enhancement of their manuscript, and I do appreciate their work. Nevertheless, major concerns have not been eradicated. Even though the main text was improved still, elementary mistakes are detectable. The language is still intricate and difficult to follow, but that might result from the Authors’ manner of writing, a rather unchangeable feature. I still have some doubts regarding statistical methods… It is not enough to mention the number of animals used only in the reviewers’ comment file – I would like the Authors to add such information to the main text. Moreover, I would like to see the normal distribution of the raw data.

To sum up, I appreciate the  Authors’ effort in both writing the original version of the manuscript and all the revisions. The work itself is interesting and worth presentation to a broader public. Nevertheless, I am not sure whether the choice of IJMS was the best option due to this journal’s standards. I do uphold my  immediate decision of rejecting this paper.

Author Response

We thank the reviewer for the comment and apologize for our limitations. We really appreciate the effort of the reviewer to improve the quality of our manuscript and we really like it, regardless of the journal decision. I, personally, thank the reviewer for the idea and comments that gain my knowledge on performing the scientific research. Here, we edit the English language again, trying to reduce the complexity of the manuscript. We also put the number of mice in the method section of the new version manuscript as following “For a short-term Ahr stimulation, total 50 mice were randomized into 5-7 mice per group.” and “Total 36 mice were randomized into 9 mice per group for WT (PBS-WT and 1,4-CQ WT) and 7 mice per group for FcgRIIb-/- mice (PBS-KO and 1,4-CQ-KO).”. Also, we demonstrate the representative of the distribution of the raw data in the figure here. For the short-term experiments, there was 8 groups (50 samples)/ time point. 

Reviewer 2 Report

The manuscript is considerably improved.

There are no major comments.

 Figures: can be improved more by distinguishing each group using specific colors throughout the manuscript

(for example, N/1,4-CQ is white, 1,4-CQ/LPS is yellow,  xx/ooo is blue..., respectively)

Author Response

We thank the reviewer for the comment and correct it accordingly.

Reviewer 3 Report

In the question #4, the author did not explain that if FcgRIIb-/- mice have higher LPS concentrations in vivo and higher enough to cause macrophage activation to worsen disease progression.

Author Response

We thank the reviewer for the comment. We agree with the reviewer in regard to an impact of the different level of LPS between WT and FcgRIIb-/- mice with long-term 1,4 CQ. Hence, we discuss more about the impact of the higher level of LPS in FcgRIIb-/- mice in the discussion of the new version manuscript as following “Also, the high LPS concentrations in serum of FcgRIIb-/- mice with long-term 1,4-CQ might be high enough to activate macrophage pro-inflammatory responses that worsen lupus disease progression. Meanwhile, endotoxemia was not detectable in WT mice with long-term 1,4-CQ possibly due to the presence of the inhibitory FcgRIIb in WT mice.”
